# Nutritional Composition of Black Soldier Fly Larvae (*Hermetia illucens* L.) and Its Potential Uses as Alternative Protein Sources in Animal Diets: A Review

**DOI:** 10.3390/insects13090831

**Published:** 2022-09-13

**Authors:** Shengyong Lu, Nittaya Taethaisong, Weerada Meethip, Jariya Surakhunthod, Boontum Sinpru, Thakun Sroichak, Pawinee Archa, Sorasak Thongpea, Siwaporn Paengkoum, Rayudika Aprilia Patindra Purba, Pramote Paengkoum

**Affiliations:** 1School of Animal Technology and Innovation, Institute of Agricultural Technology, Suranaree University of Technology, Muang, Nakhon Ratchasima 30000, Thailand or; 2Program in Agriculture, Faculty of Science and Technology, Nakhon Ratchasima Rajabhat University, Muang, Nakhon Ratchasima 30000, Thailand

**Keywords:** alternative feed ingredient, black soldier fly larva, poultry, protein, swine

## Abstract

**Simple Summary:**

With the increasing global population, the shortage of protein feed resources is becoming more and more serious, and human beings urgently need to find new protein sources to replace traditional soybean meal and fish meal. Black soldier fly larvae are rich in fatty acids, proteins and minerals. Numerous studies have shown that adding black soldier fly larvae powder to monogastric animals has no negative effects on animal growth performance, meat quality and immunity. However, black soldier flies are still subject to legal constraints and consumer acceptance when they are used as feed.

**Abstract:**

The rapidly growing population has increased demand for protein quantities and, following a shortage of plant-based feed protein sources and the prohibition of animal-based feed protein, has forced the search for new sources of protein. Therefore, humans have turned their attention to edible insects. Black soldier fly larvae (BSFL) (*Hermetia illucens* L.) are rich in nutrients such as fat, protein and high-quality amino acids and minerals, making them a good source of protein. Furthermore, BSFL are easily reared and propagated on any nutrient substrate such as plant residues, animal manure and waste, food scraps, agricultural byproducts, or straw. Although BSFL cannot completely replace soybean meal in poultry diets, supplementation of less than 20% has no negative impact on chicken growth performance, biochemical indicators and meat quality. In pig studies, although BSFL supplementation did not have any negative effect on growth performance and meat quality, the feed conversion ratio (FCR) was reduced. There is obviously less research on the feeding of BSFL in pigs than in poultry, particularly in relation to weaning piglets and fattening pigs; further research is needed on the supplementation level of sows. Moreover, it has not been found that BSFL are used in ruminants, and the next phase of research could therefore study them. The use of BSFL in animal feed presents some challenges in terms of cost, availability and legal and consumer acceptance. However, this should be considered in the context of the current shortage of protein feed and the nutritional value of BSFL, which has important research significance in animal production.

## 1. Introduction

The International Feed Industry Federation (IFIF) reports that the world population will exceed 10 billion by 2050 [1]. By then, the increased population will consume twice as much animal protein as it does today; between 2010 and 2050, consumption of pork and poultry is expected to grow by 105% and 173%, respectively [2,3], which will create enormous challenges in the production of protein feed ingredients, estimated as being more than 1.3 billion tons of dry matter [4]. Soy grains and some oilseed cakes (e.g., soybean meal, rapeseed meal, and cottonseed meal), and some algal biomass are currently the main sources of protein for ruminants and monogastric animals [5]. Global livestock feed production was estimated to be 1.104 billion tons in 2018 alone, with a total value of more than USD 400 billion [6]. Protein is the most costly and restrictive ingredient in feed formulations, and the price of traditional sources of protein has risen significantly due to yield factors and competition between humans and animals [6,7,8]. In order to meet the demand for highly nutritious animal food, the future will drive animal production systems to find new sources of high-quality and sustainable protein feed raw materials. Economic and environmental concerns must be considered in this context, while competition with plant-based human and animal food chains is reduced [9,10,11,12]. Therefore, urea can be added as a protein source in ruminant diets to improve their growth performance and milk production performance [13,14,15,16]. However, in monogastric species, ammonia concentrations in the gastrointestinal tract and the environment can cause damage to the gastrointestinal mucosa, resulting in impaired nutrient absorption, energy inefficiency, and reduced growth performance [17]. Thence, people have turned their attention to insects with high protein content.

As protein raw materials, insects such as BSFL, mealworm larvae (*Tenebrio molitor* L.), and crickets (*Orthoptera: Gryllidae*) are the focus of emerging research fronts and are already used as alternative nutrient sources for poultry and swine feed because they contain nearly 100 percent of the edible portion of protein. [18,19,20]. The feed conversion rate of BSFL is better than that of mealworms and crickets, and its survival rate and nitrogen and phosphorus composition do not change greatly with a change in diet [21]. BSFL are characterized by a high food conversion rate, short reproductive cycle, and high content of fat, protein, minerals, and vitamins [22,23]. They are also highly sustainable as they can be raised on a large scale in organic streams, and at a much lower environmental cost than traditional protein sources [24,25]. Multiple studies have shown that BSFL can be used as a food or feed source, ultimately helping to solve the global food problem. However, consumers prefer to use BSFL in animal feed rather than directly for human consumption, because people have a certain degree of psychological aversion to eating insects [26,27,28]. Diets supplemented with BSFL appear to improve growth performance and digestibility in pigs and poultry compared to other protein feeds [29,30]. Schiavone et al. [31] used whole-fat BSFL to replace 50% or 100% soybean oil, and the results showed that the growth performance, blood biochemical immune parameters, and intestinal health of broilers were not impaired. BSFL can also reduce the quality and nutrient content of pig manure with an efficiency similar to that of poultry manure, which is beneficial for improving farm hygiene [32,33], reducing pest numbers, and reducing nutrient pollution in runoff [34]. Therefore, BSFL can replace traditional protein feed ingredients such as soybean meal and fish meal. This study reviews the nutritional value of BSFL, their effects on animal performance, digestion, and immunity, and the challenges they may face when fed to animals.

## 2. BSFL (*Hermetia illucens* L.)

BSFL (*Hermetia illucens* L.; Diptera: Stratiomyidae) are a saprophytic insect that primarily feeds on organic wastes such as plant residues, animal manure, and waste, food waste, agricultural byproducts, or straw [32,35,36,37]. BSFL are an excellent candidate for human and animal protein sources, and the utilization of organic waste can help to reduce pollution [38,39,40,41]. In the process of degrading waste, BSFL converts organic waste into amino acids, peptides, proteins, oils, chitin, and vitamins, thereby controlling certain harmful bacteria (such as Salmonella and Escherichia coli) and pests, and are also used in medicine and chemical and various animal feeds (mainly pets, pigs and poultry) [38,42].

BSFL originated in the South American savannah and are widely distributed in temperate, subtropical, and tropical regions, with an optimum temperature range of 25°C to 30 °C [43,44]. Due to their lack of resistance to cold, they cannot survive in northwestern Europe and regions with temperatures below 5 °C [45]. BSFL are one of five genera in the subfamily Hermetiinae of the order Diptera [46]. The other four genera are Patagiomyia, Chaetosargus, Notohermetia, and Chaetohermetia; *Hermetia illucens* is the most widespread of all species [47]. It is a large, slender black species with three segments—head, thorax, and belly—with brownish wings and tentacles projecting from the head [48]. There are five segments on the abdomen with white spots. Males are longer than females but have smaller end genitals and wings. Females have body lengths between 12 and 20 mm and wings between 8 and 14.8 mm [48]. Their life cycle has five stages: egg, larva, pupa, pre- and adult. The larval and pupal stages are the most nutrient-rich and largely depend on the quality of food, with about 18–33% fat and 32–53% protein [49,50,51]. BSFL have a lifespan of approximately 20–22 days, with a pupa for the first 6–8 days and an adult metamorphosis for the last 14 days [47,52,53]. Adult worms have no mouth, digestive system, or stinger, pose no threat to other organisms [54] and have no affinity for the human body and fresh food. Therefore, they also do not serve as vectors for disease transmission [55].

Bioconversion rate is one of the important indicators of waste efficiency used to treat BSFL [56,57], and the biotransformation rate depends on many factors, such as the concentration of digestible nutrients, protein, fat, fiber, pH, feeding rate [37], density and water content of substrates, etc. [37,58]. The ideal moisture content is between 60% and 80%, with a lower limit of about 40% [59]. The management requirements of livestock manure around the world are getting higher and higher [60,61,62]. Supplementing BSFL in livestock feed can reduce the excretion of manure by 60% [63]. Moreover, the larvae can also decompose more than 50% of chicken manure, and convert it into high-quality amino acids, protein, and fat for animal feed, reducing the cost of breeding [64]. At the same time, BSFL nutrients are also rich in minerals and chitin and have antioxidant and immune-boosting properties (discussed in the results below).

## 3. Nutritional Value of BSFL (*Hermetia illucens* L.)

### 3.1. Regular Nutrition Facts

In animal feed, there are currently two types of BSFL: defatted and full fat, with the primary difference being in fat and saturated fatty acid content. Table 1 shows its nutritional value. The average crude protein content of BSFL was 414.7 g/kg, ranging from 216 g/kg [65] to 655 g/kg [66], which was lower than a conventional soybean meal (CSBM) (494.4 g/kg) and fish meal (675.3 g/kg) [67]. The protein content of full-fat BSFL is relatively similar, but the protein content after defatting is very different (from 216 g/kg to 655 g/kg), which may be related to the method of defatting, such as the irreversible damage to the protein caused by high temperatures. In full-fat BSFL, the fat content ranged from 294 g/kg [68] to 515.3 g/kg [69], with an average of 353.2 g/kg; these were both higher than CSBM (14 g/kg) and fish meal (103.6 g/kg) [67]. The average fat level after defatting (69.2 g/kg) was also higher than CSBM. BSFL contained ash at an average of 82.4 g/kg, ranging from 27 g/kg [45] to 132 g/kg [68], higher than CSBM (71.9 g/kg) and lower than fish meal (171.5 g/kg) [67]. The average level of crude fiber was 95.4 g/kg, ranging from 41 g/kg [45] to 213 g/kg [68], which was lower than CSBM (74.3 g/kg), but higher than fish meal (2.6 g/kg) [67]. The average content of chitin was 61.7 g/kg, ranging from 38.7 to 72.1 g/kg. The active ingredient of chitin is chitosan, which is another important polysaccharide in addition to cellulose. Chitin (linear polymer of -(1-4)*N*-acetyl-d-glucosamine units) and cellulose (linear polymer of -(1-4)d-glucopyranose units) have similar molecular structures [70]. Chitin is considered an indigestible fiber, but it can improve the immune function of animals [71,72]. The variation of crude fiber content in BSFL may be related to developmental stages, and the closer to metamorphic adults, the higher the fiber content [73]. Therefore, different research results produced different levels of chitin content.

### 3.2. Amino Acid Profile

Both defatted and full-fat BSFL have a rich amino acid profile and are thus considered a more sustainable protein source than CSBM or fish meal [77]. The amino acid profile of BSFL is shown in Table 2. The most abundant essential amino acids were leucine (average 44.6 g/kg, from 27.8 g/kg to 78.3 g/kg), lysine (average 38.8 g/kg, from 23.0 g/kg to 68.2 g/kg) and valine (average 40.1 g/kg, ranging from 28.2 g/kg to 67.9 g/kg). These three amino acid contents are higher than those of soybean meal, and even the valine content is higher than that of fish meal [67]. The least abundant essential amino acids are methionine and tryptophan, which are comparable to soybean meal and are much lower than fish meal [67]. The content of histidine ranged from 9.8 g/kg to 48 g/kg, and the content of isoleucine ranged from 17.7 g/kg to 48 g/kg, which was slightly higher than soybean meal and fish meal [67]. The content of phenylalanine ranged from 16.4 g/kg to 77.6 g/kg, and the content of threonine ranged from 16.2 g/kg to 45 g/kg, which are basically the same as soybean meal and fish meal. Arginine and histidine are lower than soybean meal and fish meal [67].

### 3.3. Fatty Acid Profile

The fatty acid content of BSFL is shown in Table 3. The most abundant saturated fatty acids (SFA) are lauric acid (C12:0), which ranges from 75 to 575. An amount of 6 g/kg, myristic acid (C14:0), which ranges from 23 to 98.7 g/kg, palmitic acid (C16:0), which ranges from 10.3 to 192.0 g/kg, and stearic acid (C18:0), which ranges from 9.8 to 69.0 g/kg. The highest content of monounsaturated fatty acids is oleic acid (c9C18:1), which ranges from 79.7 to 266.0 g/kg, palmitoleic acid (C16:1), which ranges from 10.3 to 192.0 g/kg, linoleic acid (C18:2n6), which ranges from 38.0 to 314.0 g/kg, and linolenic acid (C18:3n3), which ranges from 9.8 to 36.0 g/kg. SFA content ranges from 362.0 to 782.9 g/kg, MUFA ranges from 85.5 to 287.0 g/kg, n-6 PUFA ranges from 80.0 to 314.0 g/kg and n-3 PUFA ranges from 9.8 to 36.0 g/kg. The MUFA/SFA ratio ranges from 15.3% to 79.3% and the n-3 PUFA/n-6 PUFA ratio ranges from 8.5% to 17.9%.

### 3.4. Minerals Composition

Table 4 shows the mineral content of BSFL. BSFL are rich in minerals; calcium (Ca) is the most abundant and ranges from 1.2 g/kg to 35.7 g/kg. Copper (Cu) ranges from 0.1 g/kg to 15.0 g/kg. Iron (Fe) ranges from 0.1 g/kg to 191.0 g/kg. Magnesium (Mg) ranges from 1.0 g/kg to 3.5 g/kg. Manganese (Mn) ranges from 0.2 g/kg to 166.0 g/kg. Phosphorus (P) ranges from 1.0 g/kg to 10.3 g/kg. Potassium (K) ranges from 1.7 g/kg to 15.4 g/kg. Sodium (Na) ranges from 0.7 g/kg to 15.6 g/kg. Zinc (Zn) ranges from 0.7 g/kg to 103.0 g/kg. However, in addition to the accumulation of the above minerals, some toxic and harmful elements (such as Ba, Hg and Mo) will also bioaccumulate in BSFL [85], which will pose a challenge to the safety of feed and food production [86].

### 3.5. Different Factors of Nutritional Value of BSFL (Hermetia illucens *L.*)

The content of minerals and other nutrients in BSFL significantly varies across different studies, and the reasons may be as follows:

First, it may be that the growth stages of BSFL are different. On the 4–14th days, the crude fat content of larvae increased rapidly, and the highest level could reach 28.4%, while crude protein showed a continuous downward trend at the same developmental stage. With the development of pupa, crude fat dropped sharply to 24.2%. The maximum crude protein in adulthood is 57.6% and the fat level is 21.6% [88].

Second, in relation to the nutritional structure ingested by BSFL, the content of fat and ash fed from vegetable waste, chicken feed and kitchen waste varies greatly [23,89]. In addition, BSFL on cow dung grow at a much slower rate of individual size than on poultry feed [90,91].

Third, it may be related to the processing method. Different killing methods (such as blanching, drying, freezing, high hydrostatic pressure grinding, and asphyxiation) also had an effect on pH, ash, fat content and oxidative capacity [92,93]. The temperature and method during storage also affect the nutritional quality of BSFL [94]. Different extraction methods also have a different nutrient content; for example, the best separation of protein is through alkali extraction [95]. When the processing temperature is 25 °C, the shelf life of BSFL can reach seven months [96].

Fourth, it is related to various factors such as temperature, humidity, sunlight, moisture content, pH, etc. Humidity and temperature will obviously affect the incubation, development and lifespan of BSFL [97,98]. Temperatures between 26 and 40 °C and relative humidity between 40–70% are the ideal living conditions for BSFL [55,99]. Sunlight also affects the nutrient composition of BSFL, with black soldier flies developing best in the wavelength range between 450 and 700 nm [54,100]. When the water content in the feed matrix is 60–80%, the survival rate and growth rate of BSFL are the highest [37,101]. The growth of black soldier flies is better under alkaline conditions than under acidic conditions, and a suitable pH value is between 6–9 [102,103].

## 4. Nutrition of BSFL (*Hermetia illucens* L.) in Poultry

### 4.1. Growth Performance

The effects of BSFL on the growth performance of poultry are shown in Table 5. Replacing 50% or 100% of soybean meal in layer diets with defatted BSFL had no effect on feed intake, egg production, yolk, shell weight, and the occurrence of mortality [104]. However, Murawska et al. [105] used full-fat BSFL instead of soybean meal. When 100% was replaced, the LW of the whole stage and average daily gain (ADG) (days 1–14 and days 14–35) were reduced. The higher the replacement level, the lower the daily feed intake (DFI), FCR, carcass weight, and muscle weight. In addition, abdominal fat deposits were also increased. This may be due to the fact that BSFL are not defatted, and the 100% replacement of soybean meal results in a high proportion of saturated fatty acids in the diet. Dabbou et al. [106] found that the dietary supplementation of 10% BSFL significantly increased LW, ADG, DFI, and FCR on days 10–35, whereas it did not affect FCR on days 1–10. Similar results were obtained by Onsogo et al. [68], who indicated that supplementation of 10% BSFL in the diet increased the final weight of broilers. However, Kawaseki et al. [107] indicated that supplementation of 10% BSFL had no effect on feed intake, BW, liver weight, egg-laying rate, and egg shell weight, but increased the richness of cecal microbiota. Schiavone et al. [31] reported that supplementation of 10% BSFL had no effect on daily weight gain (DWG), DFI and FCR, live weight (LW), chilled carcass, breast, thighs, abdominal fat, liver, heart, and spleen. Studies by others have produced similar results; supplementation of BSFL in the diet of chickens did not affect the LW, ADG, DFI, FCR, chilled carcass, breast, thighs, abdominal fat, liver, heart, and spleen [68,105,108,109]. Gariglio et al. [110] found that dietary supplementation of BSFL did not affect the LW, ADG, DFI, and FCR of ducks, but reduced SW and corresponding hot carcass (HC) and chilled carcass (CC). The abdominal fat weight showed a quadratic response, the lowest corresponding to the 6% group. To sum up, the analysis of the above results, adding 10% BSFL to the diet can improve the growth performance of chickens; 100% of soybean meal had a negative impact, and adding other levels had no effect.

### 4.2. Antioxidants and Immunity

The effects of BSFL on the antioxidants and immunity of poultry are shown in Table 6. Dabbou et al. [106] found that dietary supplementation of 5%, 10%, and 15% BSFL had no effect on erythrocyte, aspartate aminotransferase (AST), creatinine, triglycerides, cholesterol, heterophiles to lymphocytes ratio (H/L), leukocyte and urea acid, but increased levels of glutathione peroxidase (GPx) and total antioxidant status (TAS). Gariglio et al. [109] also obtained similar results, finding that supplementing 3%, 6%, and 9% BSFL had no effect on the erythrocyte, heterophils, lymphocytes, basophils AST, alanine aminotransferase (ALT) and gamma-glutamyl transferase (GGT) of chickens. However, the concentration of cholesterol dropped from 30.23% to 23.86% in the 9% group compared with the control, and also reduced malondialdehyde (MDA) and nitrotyrosine levels in plasma. These data indicate that BSFL have antioxidant capacity. Schiavone et al. [31] report that even substituting 100% of BSFL had no effect on the biochemical immune performance of broilers. These results are in agreement with the findings of Loponte et al. and Bellezza Oddon et al. [108,111], who showed that supplementation of 5% BSFL had no effect on the biochemical immune performance of chickens. However, the findings of Marono et al. [112] were slightly different from the above results. Supplementation of 100% BSFL increased concentrations of cholesterol, globulin, and triglycerides, but had no effect on other immune parameters. Gariglio et al. [109] showed that the supplementation of BSFL in a duck’s diet did not affect the levels of plasma AST, ALT, GGT, GPx, TAS, and methylglyoxal (MG), while MDA decreased linearly with the increase in BSFL. The above findings on the ratio of globulin to albumin provide an overall increase in circulating immunoglobulins, demonstrating that the supplementation of BSFL in poultry diets has a greater resistance to disease and better immune function, at least without adverse effects or influences [22].

### 4.3. Meat Quality and Nutritional Content

The effects of BSFL on the meat quality and nutritional content of poultry are shown in Table 7. The study by Cullere et al. [113] using BSFL to replace soybean oil found that it had no effect on moisture, protein, lipid and ash in meat, cholesterol and TBARs; pH, L*, a*, b*, thawing loss and cooking loss had no effect either. However, it did increase ratio levels of SFA and n-6/n-3, and reduce MUFA, PUFA and n-3 PUFA. Similar results were obtained by Moula et al. [114], who reported that the FA content of added BSFL was 36.7% higher than that of the control group, which was 33.8%. Murawska et al. [105] obtained different research results, finding that BSFL replacing 50% soybean meal can increase muscle b* and replacing more than 75% will increase muscle pH, but there is no effect on aroma intensity or desirability. The study by Schiavone et al. [115] showed that the dietary supplementation of 10% BSFL increased a* and the protein content of muscle; at 15% supplementation, b* and moisture in the muscle were reduced, and the ratio of FA/PUFA gradually increased with increasing levels of supplementation. The higher red color may be due to the accumulation of pigments in the insect meal in the meat, while the yellow color may be related to the corn gluten content of the insect meal’s diet. Muscle FA content is directly related to the type of fatty acid in BSFL. Cullere et al. [116] found that supplementation of 10% and 15% BSFL increased the content of aspartic acid, glutamic acid, alanine, serine, tyrosine, and threonine in muscle. This shows that the amino acids in BSFL can be absorbed and preserved by poultry. Gariglio et al. [110] found that black soldier flies did not affect pH 24 and the color of the chest and thigh muscles. BSFL did not affect the SFA content of breast meat, and the levels of lauric acid (C12:0) and myristic acid (C14:0) increased with the increase in BSFL; α-linolenic acid (C18:3 n-3) in breast meat also significantly increased. Linear quadratic decline and the ratio of ∑n-6/∑n-3 of breast and thigh meat decreased linearly, with the lowest in the 9% group.

## 5. Nutrition of BSFL (*Hermetia illucens* L.) in Swine

### 5.1. Growth Performance

Table 8 shows the effect of BSFL on pig growth performance. The use of BSFL to replace fish meal in part or entirely increased the average daily gain of finishing pigs. The final body weight, fasted weight, and carcass weight were significantly higher in the 50% and 100% supplemented groups than in the unsupplemented and 25% supplemented groups. The FCR of the BSFL-supplemented diet, however, was significantly lower than that of the unsupplemented group [117]. Driemeyer [118] supplemented 3.5% BSFL in piglet diets without affecting ADG, DMI, and digestibility, but reduced FCR. Crosbie et al. [77] found that on the seventh day, the weight of pigs fed the 25% BSFL diet was higher than that of the pigs fed the 50% diet, and the BSFL had no effect on body weight at other stages, decreased ADMI and G: F on days 14–21. Biasato et al. [119] found that BSFL did not affect the digestibility of BW, WG, ADG, ADFI, FCR, and nutrients of piglets. A similar result was obtained by Ipena et al. [120], who found that supplementation of BSFL did not affect average daily gain and the final body weight of piglets, nor did they affect their feed efficiency and energy efficiency, and had an effect on piglet diarrhea. Yu et al. [121] showed that supplementing 4% BSFL increased the abundance of lactobacillus, pseudo-butyric vibrio, rosella, and faecalibacterium in the swine intestinal tract, while the abundance of streptococcus decreased. Supplementation at a level of 8% increased the number of Clostridium cluster XIVa. The above results indicate that BSFL not only had no negative effects on the feed intake, digestion and growth performance of pigs, but also promoted the development of gut microbes while reducing FCR.

### 5.2. Antioxidants, Immunity, and Meat Quality

Since there is not as much research on BSFL in swine as there is in poultry, we counted blood indicators and meat quality indicators together. The effects of BSFL on the antioxidants, immunity, and meat quality of pigs are shown in Table 9. Research by Biasato et al. [119] showed that dietary BSFL supplementation had no significant effect on the animals’ blood chemistry profile or serum protein, except that monocytes and neutrophils exhibited linear and quadratic responses, respectively, to increases in the dietary levels of BSFL. Research by Chia et al. [117] showed that replacing 50% of fish meal with BSFL increases the muscle protein content, while BSFL at any level increases muscle total fat and organic matter (OM) content, but has no effect on muscle water content. Altmann et al. [122] found that the dietary supplementation of BSFL had very little effect on pork quality parameters and sensory attributes. All physicochemical parameters, except backfat (L*), were not affected by diet. In addition, BSFL increased the content of polyunsaturated fatty acids (PUFA), GLA and linoleic acid (C18:2), and lauric acid (C12:0) was five times higher. This may be related to the content of saturated fatty acids in BSFL.

## 6. Challenges of Using BSFL (*Hermetia illucens* L.)

### 6.1. Security and Legal Issues

At present, the main obstacles to the large-scale use of BSFL in animal feed are the legal regulations related to food safety [123]. Despite the existence of substantial funding for research on BSFL, restrictive European laws regarding the use of insects in animal feed remain a major challenge for the development of insect production units [124]. The Food and Agriculture Organization of the United Nations’ global food regulations establishes safe conditions for BSFL to meet the production of insects for breeding and pet animal feed [125]. That is: “These should not be pathogenic or have other adverse effects on plant, animal or human health; they should not be considered carriers of plant, animal or human pathogens and should not be protected or defined as invasive alien species”. They also place restrictions on the substrates of BSFL’s nutritional source: the substrate must contain “products of non-animal origin” or a limited set of animal products, including fish meal, refined fats, blood and gelatin from non-ruminant animals, milk, eggs, honey, etc. Meat, manure, “catering waste” and “other waste” are explicitly excluded [126]. Although the U.S. Food and Drug Administration (FDA) has an official Memorandum of Understanding with the American Association of Feed Control Officials (AAFCO) for all regulations governing animal feed, it is also only sold as a human novelty food and is not expressly identified as food [126,127].

### 6.2. Consumer Acceptance

Consumer acceptance of BSFL-fed animal meat products should also be considered. There is general support for the idea of using insects in animal feed, especially in fish and poultry feed. A questionnaire found that two-thirds of study participants were willing to accept the use of insects in animal feed [128]. Sogari et al. [129] also obtained similar findings, with the most accepted being the use of insects in fish feed. This shows that consumers still have certain doubts about insect-fed animals, and efforts need to be made to improve consumers’ perceptions of insect-fed animal meat products. In addition, there are also findings of direct human consumption of BSFL. In the Malaysian province of Sabah on the island of Borneo, more than 60 species of insects are eaten, primarily by certain groups of the indigenous Kadazan Dusun people. BSFL are one of them and are eaten raw with a locally brewed fermented beverage called tapai [130]. In addition, the strong relationship between dietary SFA and coronary heart disease (CHD) risk may also be one of the reasons why consumers do not want to consume BSFL directly [131].

### 6.3. Production and Price

At present, there is no unified standard for the production and subsequent processing of BSFL. Due to the small scale of production equipment, low yield, and low efficiency, there is only enough BSFL for consumers who are early adopters and will use them in small amounts for scientific research feed [132]. Mass production is still a long way off. In this case, the availability and price of BSFL are not competitive with soybean meal and fish meal, resulting in BSFL not being widely used in poultry or swine diets [124].

## 7. Conclusions

In conclusion, the high protein content of BSFL has great potential as a protein source in poultry and swine diets, although there is no information on its application in ruminant diets. According to the results of its application in monogastric animals and ruminants with a special rumen function, BSFL has a lot of room for future development in ruminant feed. Although studies on the effects of BSFL on growth performance, antioxidants, immunity, and meat quality in pigs and poultry have been inconsistent (as to whether they have caused improvement or not), there have been no reports of the negative effects of BSFL on pigs or poultry. This shows that BSFL can replace soybean meal or fish meal as a feed protein source. Due to the relatively short application time of BSFL in animal feed, the development of related laws is relatively backward. It is necessary to formulate standard production and processing procedures and improve relevant laws to promote the production of BSFL and reduce prices. Furthermore, consumer acceptance of meat products from insect-fed animals is a challenge to overcome.

## Figures and Tables

**Table 1 insects-13-00831-t001:** Regular nutrition facts of BSFL (g kg^−1^ dry matter basis).

BSFL										CSBM	FM
Type	FF	DF	DF	FF	FF	FF	FF	FF	DF		
Crude protein	431.0	655.0	216.0	411.0	439.0	350.0	401.0	275.4	554.2	494.4	675.3
Crude fat	386.0	46.0	63.0	301.0	294.0	298.0	325.0	515.3	98.5	14.0	103.6
Crude fiber	41.0		70.0		213.0	79.0			74.0	74.3	2.6
Ash	27.0	93.0	93.0	93.0	132.0	53.0	104.0	65.9	81.0	71.9	171.5
Chitin	67.0	69.0						38.7	72.1		
References	[23]	[18]	[65]	[74]	[68]	[75]	[76]	[69]	[69]	[67]	[67]

CSBM = conventional soybean meal, FM = fish meal, FF = full-fat, DF = defatted.

**Table 2 insects-13-00831-t002:** Amino acid composition of BSFL (g kg^−1^ dry matter basis).

BSFL								CSBM	FM
Indispensable amino acids									
Type	FF	DF	FF	FF	FF	FF	FF		
Arginine	19.9	20.7	21.1	54.7	62.0	21.9	18.7	35.7	41.0
Histidine	13.8	16.3	13.5	32.5	48.0	9.8	13.7	14.2	15.4
Isoleucine	19.1	24.0	17.7	47.3	48.0	19.1	20.6	22.1	27.3
Leucine	30.6	36.7	27.8	78.3	77.0	32.1	29.4	38.6	47.7
Lysine	23.0	25.2	28.1	68.2	74.0	27.2	25.9	31.1	48.7
Methionine	7.1	8.56	8.0	21.2	6.0	6.0	7.1	.6.8	18.5
Phenylalanine	16.4	21.8	16.4	77.6	62.0	18.3	18.7	25.5	26.4
Threonine	16.2	21.8	16.3	44.3	45.0	26.5	16.7	19.8	27.5
Tryptophan	5.4					5.6	6.3	6.6	6.7
Valine	28.2	34.5	25.0	67.9	67.0	28.7	28.8	21.7	32.7
Dispensable amino acids									
Alanine	27.8	43.7	25.6	82.1	62.0		26.6	21.6	41.9
Aspartic acid	36.9	48.8	38.7	73.0	103.0		35.6	55.0	57.7
Cysteine	2.2	0.2	3.5	7.6	5.0	4.2	3.2	7.7	6.5
Glycine	25.2	30.3	24.6	61.5	54.0	26.8	24.8	21.3	50.3
Glutamic acid	45.8	63.7	46.1	131.0	102.0		38.4	88.6	84.1
Proline	25.1	32.7	23.6	66.8	62.0		23.1	27.4	30.8
Serine	15.9	26.8	17.6	48.8	41.0	19.2	15.2	24.1	25.9
Tyrosine		34.1		67.1	60.0	26.5	26.9	15.5	20.1
References	[23]	[18]	[68]	[75]	[78]	[79]	[80]	[67]	[67]

CSBM = conventional soybean meal, DF = defatted, FM = fish meal, FF = full fat.

**Table 3 insects-13-00831-t003:** Fatty acid composition of BSFL (g kg^−1^ dry matter basis).

Type	FF	FF	FF	FF	FF	FF
C10:0	20.3		8.6	14.3		8.6
C12:0	575.6	75.0	459.7	526	468.6	407.9
C14:0	71.4	23.0	87	85.4	98.7	65.6
C15:0			1.5		143.8	1.3
C16:0	10.3	192.0	122.1	109	143.8	162.7
C18:0	9.8	69.0	25.3	15.3	17.9	14.3
SFA	782.9	362.0	707.2	750.0	742.4	664.2
C16:1	33.4	8.0	19.1	19.8	27.8	23.6
c9C18:1	79.7	266.0	112.4	61.6	77.3	182.4
c11C18:1	1.2			2.4		
MUFA	119.9	287.0	134.1	85.5	115.8	218.8
C18:2n-6	78.3	314.0	38.0	116.0	127.7	100.7
n-6 PUFA	80.0	314.0	142.2	119.0	106.0	100.9
C18:3n-3	11.0	36.0	16.5	10.1	9.8	16.0
C18:4n-3	0.5					
C20:5n-3	2.3					0.2
C22:6n-3	0.1					
n-3 PUFA	14.3	36.0	16.5	10.1	9.8	16.2
MUFA /SFA, %	15.3	79.3	18.9	11.4	15.6	32.9
n-3 PUFA/ n-6 PUFA, %	17.9	11.5	11.6	8.5	9.0	16.1
References	[23]	[81]	[82]	[75]	[83]	[84]

DF = defatted, FF = full fat, MUFA = monounsaturated fatty acid, PUFA = polyunsaturated fatty acids, SFA = saturated fatty acid.

**Table 4 insects-13-00831-t004:** Mineral compositions of BSFL (g kg^−1^ dry matter basis).

Type	FF	DF	FF	FF	FF
Calcium (Ca)	1.2	13.0	1.9.0	34.6	35.7
Copper (Cu)	0.1	15.0	0.6	10.7	0.7
Iron (Fe)	0.1	125.0	2.1	191.0	14.0
Magnesium (Mg)	2.1	3.0	1.0	3.5	3.4
Manganese (Mn)	0.2	45.0	0.3	166.0	33.5
Phosphorus (P)	4.1	8.0	1.0	10.3	7.0
Potassium (K)	6.0	11.0	1.7	15.4	9.2
Sodium (Na)	0.7	5.0	3.3	1.7	15.6
Zinc (Zn)	0.7	90.0	0.9	103.0	9.0
References	[23]	[65]	[74]	[79]	[87]

DF = defatted, FF = full fat.

**Table 5 insects-13-00831-t005:** Effects of BSFL on the growth performance of poultry.

References	Breed	Type	Level, %	Age	Performance
[104]	Lohmann Selected Leghorn	DF	50, 100	128 days	No difference in feed intake, egg production, yolk and shell weight. No mortality occurred.
[107]	Laying Hens	FF	10	168 days	No difference in feed intake, BW, liver weight, and the egg-laying rate. No difference in egg yolk weight, egg shell weight, egg shell strength, and Haugh unit. Increases the richness of cecal microbiota.
[106]	Ross 308	DF	5, 10, 15	1–35 days	No difference in FCR (days 1–10). Increases the LW, ADG, DFI and FCR (days 10–35).
[31]	Ross 308	FF	50, 100	21–48 days	No difference in DWG, DFI and FCR.
					No difference in LW, chilled carcass, breast, thighs, abdominal fat, liver, heart and spleen.
[68]	Cobb 500	FF	5, 10, 15	1–79 days	No difference in final weight, DFI and FCR (days 7–28). Increases the final weight (days 28–79). No effects on breast meat weight, abdominal fat content, and internal organ (liver, heart, gizzard, and spleen) weights.
[108]	Ross 308	FF	5	1–39 days	No difference in final weight DFI and FCR. No difference in slaughtering performance.
[105]	Ross 308	FF	50, 75, 100	1–42 days	Reduces the LW of the whole stage. Decreased ADG (days 1–14, days 14–35). Reduces DFI and FCR in the 100% groups. Reduces carcass weight and muscle weight. Increases abdominal fat deposits. Increases carcass fat content and the proportion of viscera to the total weight.
[109]	R71 L	FF	3, 6, 9	3–50 days	No difference in LW, ADG, DFI and FCR. Reduces mortality (from 4.16% to 2.08%).
[110]	Canedins R71 L White	DF	3, 6, 9	3–50 days	No difference in LW, ADG, DFI and FCR. SW, and consequently the HC and CC weights, and abdominal fat showed a quadratic response with the minimum corresponding to the 6% group.

ADG = average daily gain, BW = body weight, CC = chilled carcass, DF = defatted, DFI = daily feed intake, DWG = daily weight gain, FCR = feed conversion ratio, FF = full-fat, HC = hot carcass, LW = live weight, SW = slaughter weight.

**Table 6 insects-13-00831-t006:** Effects of BSFL on the antioxidants and immunity of poultry.

References	Breed	Type	Level, %	Age	Performance
[106]	Ross 308	DF	5, 10, 15	1–35 days	No difference in erythrocyte, AST, creatinine, triglycerides, cholesterol, H/L, leukocyte, and uric acid.
					Increases the activity of GPx and TAS.
[31]	Ross 308	FF	50, 100	21–48 days	No difference in GGT, H/L, ALT, AST, cholesterol, creatinine, erythrocyte, iron, leukocyte, magnesium, phosphorus, total protein, triglycerides, and uric acid.
[108]	Ross 308	FF	5	1–39 days	No difference in erythrocyte, ALT, uric acid, Albumin, AST, cholesterol, creatinine. GGT, H/L, HDL, LDL, leukocyte, total protein and triglycerides.
[109]	R71 L White	FF	3, 6, 9	3–50 days	No difference in erythrocyte, heterophils, lymphocytes, basophils, eosinophils, H/L, leukocyte, monocytes, and total protein. No difference in AST, ALT and GGT. Reduces triglycerides and cholesterol (from 30.23% to 23.86%). No difference in GPx, TAS and MG. Reduces MDA and nitrotyrosine.
[112]	Lohman Brown Classic laying hens	FF	100	21–45 weeks	Increases cholesterol, globulin and triglycerides. No difference in erythrocyte, eosinophils, basophils, H/L, heterophils, leukocyte, lymphocytes, monocytes and total protein. Reduces albumin.
[111]	Alectoris barbara	FF	25, 50	7–28 days	Reduces albumin and BUN. No difference in erythrocyte, AST, ALT, total protein and H/L.
[109]	Canedins R71 L White	DF	3, 6, 9	3–50 days	No difference in AST, ALT and GGT. No difference in GPx, TAS and MG. Reduces MDA.

ALT = alanine aminotransferase, AST = aspartate amino transferase, BUN = blood urea nitrogen, BW = body weight, DF = defatted, FF = full-fat, GGT = gamma-glutamyl transferase, GPx = glutathione peroxidase, H/ L = heterophiles to lymphocytes ratio, HDL = high-density lipoproteins, LDL = low-density lipoproteins, MDA = malondialdehyde, MG = methylglyoxal, TAS = total antioxidant status.

**Table 7 insects-13-00831-t007:** Effects of BSFL on the meat quality and nutritional content of poultry.

References	Breed	Type	Level, %	Age	Performance
[113]	Ross 708	FF	50, 100	21–48 days	No difference in moisture, protein, lipids and ash of meat. No difference in cholesterol and TBARs. No difference in pH, L*, a*, b*, thawing loss and cooking loss.
					Increases the total SFA and n-6/n-3 ratio. Decreases MUFA, PUFA and n-3 PUFA.
[105]	Ross 308	FF	50, 75, 100	1–42 days	Increases pH, a* and b*. Reduces cooking losses. No difference in aroma intensity or desirability.
[114]	Ardennaise	FF	2	30–80 days	Increases total FA (BSF 36.7%, control 33.8%) and C20: 4ω6. No difference in ω6/ω3 ratio. No difference in the protein content of meat (control 24%, BSF 26%).
[116]	Quails	FF	10, 15	10–28 days	Increases alanine, serine, aspartic acid, glutamic acid, tyrosine and threonine. Increases the SFA (C10: 0, C12: 0, C14: 0, C16: 0 and C20: 0) and n-6/n-3 ratio.
					Decreases PUFA, especially n-3.
[115]	Ross 308	FF	5, 10, 15	1–35 days	Increases a*, protein and the FA/ PUFA ratio. Reduces b* and moisture.
[110]	Canedins R71 L White	DF	3, 6, 9	3–50 days	No difference in pH 24 and the color of the breast and thigh muscles. Increases C12: 0, C14: 0 and α- linolenic acid. No difference in SFA brisket. Decreases C18:3 n-3, the lowest in the 9% group, the ratio of ∑n-6/∑n-3 decreases linearly.

a* = red/green value, b* = blue/yellow value, DF = defatted, FA = fatty acid, FF = full fat, L* = lightness, MUFA = monounsaturated fatty acid, PUFA = polyunsaturated fatty acids, SFA = saturated fatty acid, TBARs = thiobarbituric acid reactant. ∑n-6/∑ n-3 = ∑PUFA n-6/∑PUFA n-3 ratio.

**Table 8 insects-13-00831-t008:** Effects of BSFL on the growth performance of swine.

References	Phase	Type	Level, %	Age	Performance
[117]	Weaning swine	FF	25, 50, 75, 100	14 weeks	Increases ADG, FBW, fasted weight and carcass weight. Reduces FCR.
[118]	Weaned piglets	FF	3.5	10–28 days	No difference in ADG, DMI and digestibility.
[77]	Weaned piglets	FF	25, 50	7–21 days	Increases body weight at day 7. Reduces ADFI and G: F at days 14–21.
[119]	Weaned piglets	DF	5, 10	1–61 days	No difference in ADG, ADFI, BW, WG and FCR.
[120]	Weaned piglets	FF	D 21-25, 75g/day, D 25-35, 150g/day	24–35 days	No difference in ADG, final body weight, feed efficiency and energy efficiency.
[121]	Fattening pig	FF	4, 8	Initial weight 76 kg, raised for 46 days.	Increased the abundance of lactobacillus, pseudo-butyric vibrio, Rosella, and faecalibacterium of the 4% group. Decreased the abundance of streptococcus. Increased the number of Clostridium cluster XIVa of the 8% group.

ADFI = average daily feed intake, ADG = average daily gain, BW = body weight, D = days, DMI = dry matter intake, FBW = final body weight, FCR = feed conversion ratio, G: F = gain: feed, WG = weight gain.

**Table 9 insects-13-00831-t009:** Effects of BSFL on the antioxidant, immunity, and meat quality of swine.

References	Phase	Type	Level, %	Age	Performance
[119]	Weaned piglets	DF	5, 10	1–61 days	No difference in white blood cells, lymphocytes, monocytes, neutrophils, eosinophils, basophils, red blood cells, MCV, hematocrit, MCH, GOT, GPT, ALP, total protein, and triglycerides.
					No difference in albumin, alpha globulin, beta globulin, beta globulin and gamma globulin.
[122]	Fattening pig	FF	50, 75, 100	22–75 kg	Reduced back fat (L*) and all physicochemical parameters were not affected. Increases the PUFA, GLA and C18:2; C12:0 was five times higher.
[117]	Weaning swine	DF	25, 50, 75, 100	14 weeks	Increases protein, lipids and OM of meat. No difference in moisture.

ALP = alkaline phosphatase, GLA = gamma linolenic acid, GOT = glutamic oxaloacetic transaminase, GPT = glutamate-pyruvate transaminase, L* = lightness, MCH = mean corpuscular hemoglobin, MCV = mean corpuscular volume, OM = organic matter, PUFA = polyunsaturated fatty acids.

## Data Availability

All other material is from published literature referenced in the reference list.

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
