# Peer review of "Nutritional Composition of Black Soldier Fly Larvae (Hermetia illucens L.) and Its Potential Uses as Alternative Protein Sources in Animal Diets: A Review"

_insects, 2022, doi:10.3390/insects13090831_

Round 1

Reviewer 1 Report

This review is presenting the potential application of BSF as an alternative protein source in animal diets. Actually, this is an interesting hot topic nowadays, especially with the COVID-19 & climate change crisis effect 

The potential use of BSF as animal feed and human food is pushed by the food security deviation requirement. I see that this review will help to decrease the gap in knowledge of this promising industry 

However, I have some comments to improve the quality of this review  as following but not least these items

1. the authors should mention clearly the novelty of this paper as I see a lot of reviews take the same concept, for example, but not least 

Review of Black Soldier Fly (Hermetia illucens) as Animal Feed and Human Food

 https://www.ncbi.nlm.nih.gov/pmc/articles/PMC5664030/

Nutritional value of the black soldier fly (Hermetia illucens L.) and its suitability as animal feed – a review

https://avingstan.com/wordpress/wp-content/uploads/2019/08/Barragan-Fonseca-et-al-2017-Nutritional-value.pdf

2. All  Scientific names should write in italic form for example Hermetia illucens should be Hermetia illucens 

3. the information in table 1 is not required a table, I recommend removing it 

4. Table 2 the information regarding the FF, DF should be mentioned in the table legend for example why the FF is mentioned more than one time and what the difference between FF & FF type to make a difference in the protein content, the same is for table 3 , 4 &5

5. Table 4, why the authors did not compare BSF and CSBM & FM

I see that there are a lot of studies regarding this topic, please take this comment into your action the same for table 5

6. the part of production and price plus  Consumer Acceptance should be enriched with more recent studies to evaluate the target of this review 

7. the English language should be revised by native advisor

Reviewer 2 Report

“Nutritional Composition of Black Soldier Fly Larva (Hermetia illucens L.) and Potential Uses as Alternative Protein Sources in animal Diets– A Review” (insects-1878490) firstly introduced the classification and nutritional values of BSFL in detail. And then the application status of BSFL to partly replace soybean meal for feeding of chicken and pigs and the influence on their growth performance and meat quality were summarized. The challenges in terms of cost, availability, legal and consumer acceptance that BSFL faced were analyzed and the solution was also forecasted. Totally speaking, the structure of this article was reasonable and the content was relatively substantial. However, some defect needs to be modified to make this article become better.

(1) The abbreviations that occurred on the first time need to be described with the full title, such as “FCR” in the “Abstract” part. The other abbreviations should also be checked.

(2) There are some other usual animals like duck, goose etc. belong to poultry. Does the BSFL have been applied for their feeding? The author needs to add this relative content in the application of BSFL for the poultry diet. 

(3) The part “3. Nutritional value of BSFL” and “4. Different factors of nutritional value of BSFL (Hermetia illucens L.) are suggested to be combined to one section.

(4) Due to usually belong to omnivorous animals, BSFL may replace part traditional diets for the feeding of chicken and pigs. However, usually belong to herbivore, whose main diet is grass. The propotion of their diet that can be substituted may be little. Hence, if it is meaningful for the application of BSFL in ruminant?

(5) In the part of “Conclusion”, more information about the prospective investigation during the BSFL application in poultry and swine diets and the potential solution for the challenge of BSFL application should be discussed. The relative content can also be added in the corresponding part.

(6) The typical substitution process of BSFL in poultry and swine diet and the typical feeding procedure after the addition of BSFL need to be introduced and added in the corresponding part of “5. Nutrition of BSFL (Hermetia illucens L.) in poultry” and “6. Nutrition of BSFL (Hermetia illucens L.) in swine”.

Reviewer 3 Report

This paper provides a good summary of the nutritional composition of BSFL and the current status of their application in animal diets. On this basis, the challenges faced by the application of BSFL were discussed in terms of safety, legislation, consumer acceptance, and cost. And it is proposed that in order to promote the application of BSFL in animal diets, it is necessary to formulate standard production and processing procedures, improve relevant laws, reduce prices, and improve consumers' cognition. In general, the structure of this paper is reasonable and the coverage is relatively complete, but some discussions need to be strengthened. Here are the specific comments:

Line 31: "FCR", first appearance should use full spell

Line 62: "Larvae of mealworm" "larvae of mealworm"

Line 64-66: "Because the protein they contain is edible for humans and animals, and the edible portion percentage is close to 100%"

This part of expression could be better.

Line 77: BSF or BSFL?

"that"  "the"

Line 91-94: This sentence seems confusing. What are used in medicine, chemistry and various animal feeds (mainly pets, swine and poultry)? organic waste?

Line 97-98:  "Due to it lacks resistance to cold, so it cannot survive in northwestern Europe and regions with temperatures below 5°C [35]." 

"Due to its lack of resistance to cold, it cannot survive in northwestern Europe and regions with temperatures below 5°C [35]."

Line 113-114: "The biotransformation rate is an important indicator of the efficiency of the waste used for the treatment of BSFL" It's better to rephrase this sentence.

Line 116: "water content of larvae"  "water content of substrates"

Line 137, 255, Table 8: "Ash"  "ash"

Line 149, 177, 198, 208: "kg-1"  "kg-1"

Line 152-153:Please add reference.

Line 153: SBM? CSBM?

Line 158: "Methionine and Tryptophan"  "methionine and tryptophan"

Line 201: As far as I know, BSFL accumulates many metals such as Cu, Fe and Zn during the conversion process. This not only affects the mineral composition of BSFL, but also poses environmental risks for subsequent utilization. I think it is necessary to mention it where appropriate in the text.

Proc, K., Bulak, P., Wiacek, D. and Bieganowski, A. 2020. Hermetia illucens exhibits bioaccumulative potential for 15 different elements - Implications for feed and food production. Science of the Total Environment 723: 138125. https://doi.org/10.1016/j.scitotenv.2020.138125

Wu, N., Wang, X.B., Xu, X.Y., Cai, R.J., Xie, S.Y. 2020. Effects of heavy metals on the bioaccumulation, excretion and gut microbiome of black soldier fly larvae (Hermetia illucens). Ecotox. Environ. Safe., 192. https://doi.org/10.1016/j.ecoenv.2020.110323

Line 211-229: Section 4

Whether the feeding conditions (temperature, humidity, substrate moisture content, inoculation density, light) of BSFL will affect their nutritional components, if so, please supplement.

Line 232, 264: "...is shown in Table 6 (7)..."

Line 235-237: "LW", "ADG", "DFI", first appearance should use full spell

Line 267: " But can increase levels of GPx and TAS of blood". Better rewrite this sentence

Line 366: "However"  "however"

Line 379-380: " At present, the first problem that BSFL want to use in animal feed on a large scale is the legal provisions related to food safety"

It's better to rephrase this sentence.